# An Adaptive Multi-Modal Control Strategy to Attenuate the Limb Position Effect in Myoelectric Pattern Recognition

**DOI:** 10.3390/s21217404

**Published:** 2021-11-07

**Authors:** Veronika Spieker, Amartya Ganguly, Sami Haddadin, Cristina Piazza

**Affiliations:** 1Munich Institute of Robotics and Machine Intelligence, Technical University of Munich, 80797 Munich, Germany; v.spieker@tum.de (V.S.); haddadin@tum.de (S.H.); cristina.piazza@tum.de (C.P.); 2Department of Informatics, Technical University of Munich, 85748 Garching bei München, Germany

**Keywords:** upper-limb prostheses, myoelectric control, pattern recognition, limb effect, linear discriminant analysis, multi-modal control

## Abstract

Over the last few decades, pattern recognition algorithms have shown promising results in the field of upper limb prostheses myoelectric control and are now gradually being incorporated in commercial devices. A widely used approach is based on a classifier which assigns a specific input value to a selected hand motion. While this method guarantees good performance and robustness within each class, it still shows limitations in adapting to different conditions encountered in real-world applications, such as changes in limb position or external loads. This paper proposes an adaptive method based on a pattern recognition classifier that takes advantage of an augmented dataset—i.e., representing variations in limb position or external loads—to selectively adapt to underrepresented variations. The proposed method was evaluated using a series of target achievement control tests with ten able-bodied volunteers. Results indicated a higher median completion rate >3.33% for the adapted algorithm compared to a classical pattern recognition classifier used as a baseline model. Subject-specific performance showed the potential for improved control after adaptation and a ≤13% completion rate; and in many instances, the adapted points were able to provide new information within classes. These preliminary results show the potential of the proposed method and encourage further development.

## 1. Introduction

Upper limb prostheses aim to restore the lost functionalities and autonomy in daily living tasks, work and social activities. Current commercial solutions range from simple body-powered hooks to advanced poly-articulated hands, typically controlled myoelectrically [1]. Most of the latter are designed to match the appearance and functionalities of the human hand, and allow one to perform several hand motions through coordinated activation of independently motorized fingers [2,3,4]. These dexterous bionic devices use a pair of surface electromyography (sEMG) sensors to control one degree of freedom (DOF) at a time and adopt switching techniques via muscle co-activation, a mobile application and short-range proximity sensors to select the desired grip pattern [5]. Despite their potential, many individuals still consider such sophisticated myoelectric devices unreliable and difficult to control [6,7], which has drawn attention to the large discrepancy between the solutions available and users’ real needs.

To overcome these limitations, many research groups are currently investigating innovative solutions and control techniques to increase the robustness of bionic prostheses [1,8,9]. From the control point of view, several successful approaches have been proposed in recent decades, such as linear regression [10,11] and pattern recognition [12] methods. The latter have been extensively validated in laboratory environments using different techniques [13,14,15,16] and are currently being incorporated into commercial devices [17,18].

A widely used method is based on a classifier [19,20] that collects training data through a supervised calibration procedure and maps the EMG activation generated by the user to control commands for the prosthetic device. Generally, the training data are recorded “statically”—i.e., with the arm in a single position parallel to the ground [21,22]—which ensures high robustness. The performance of a classifier is evaluated by calculating the classification accuracy, which is the ability of the algorithm to correctly interpret the movements of the users. Despite promising results which show offline accuracy of more than 95% for the classification of up to 10 different motions [12], their performance and reliability in real-world environments is still limited. External factors such as sweat, fatigue, electrode shift [23,24,25] and changes in limb position and external loads [26,27,28,29] result in significant variations in EMG signals over time, which affect real-time performance. The classic approach to mitigating the adverse effects of external factors consists of adopting a dynamic training, i.e., asking the subject to actively move the arm during the data collection [30]. This approach has prompted the development of promising methods based on the collection of multiple sets of data in different pre-defined arm orientations [31] and methods that use high-density EMG arrays [32]. Both have shown significant improvements in terms of real-time control. Nevertheless, these result in increased calibration time and complexity [33,34,35], making them infeasible in real-life scenarios [36].

Previous studies have proposed a wide range of control methods, such as advanced feature extraction and classifier structures [37,38], and combinations of EMG and inertial sensors [31,33,39,40] to track arm posture. Adaptive training methods [35], based on the inclusion of additional data in real-time, can be considered as a viable direction for a robust and clinically feasible control systems. Adaptation introduced using supervised datasets [41,42,43] ensures improved classification rates, but still requires frequent retraining sessions. This limitation can be overcome with an unsupervised approach, i.e., an adaptive training method where the intended class is unknown [44,45,46]. However, this approach provides only a small reduction in error, and to the best of our knowledge, has not yet been investigated regarding the combined influence of limb pose and external load variation.

In this paper, we present an adaptive method to improve the robustness of a myoelectric pattern recognition classifier without increasing the calibration time. The proposed method utilizes an inertial measurement unit (IMU) as an additional input to the classifier, to continuously detect variations in limb position. After system calibration, where the training data are dynamically recorded, this approach incorporates adaptive unsupervised data acquisition, to include variations in muscle activation due to changes in external loads and/or limb position/orientation. This unsupervised dataset allows one to selectively adapt underrepresented external variations that are often encountered during activities of daily living which are not pre-defined. A preliminary validation of this approach was conducted in a virtual environment using the target achievement control (TAC) test [47].

## 2. Materials and Methods

### 2.1. Multi-Modal Adaptive Algorithm

The proposed algorithm was designed to improve classical myoelectric pattern recognition prosthesis control with the inclusion of additional kinematic information. Rather than extending the calibration time to generate a large labeled dataset depicting all variations of external factors, this technique was based on unsupervised adaptive data acquisition over time. The performance of an initial calibration model was evaluated and provided information about variations in hand motions and limb poses—in other words, states in need of adaptation. Unlabeled data corresponding to muscle activation detected during activities of daily living served as the source for the model adaptation. Figure 1 shows an overview of the proposed multi-modal adaptive learner. It demonstrates a data fusion strategy to introduce information about the current limb pose, a performance evaluation method to identify underrepresented states of the algorithm and an adaptation scheme to selectively include unlabeled data points for a new classification model.

#### 2.1.1. Data Fusion

Information regarding the arm position and orientation were collected using an IMU, and then integrated in the proposed algorithm in two ways. First, it was included in the feature set for the generation of hand motion classification labels, and then an initial calibration model was trained on parts of the fused feature set, which served as the reference model.

Then, limb pose labels were generated by taking advantage of the orientation data. Conversion of the provided quaternions into Euler angles allowed continuous knowledge of the arm’s rotational position with reference to a global axis, as presented in Figure 2. Under the assumption that only the subject’s limb moves and the rest of the body is fixed relative to the global reference system, the three dimensional rotations were interpreted as follows:*X* rotation, as the orientation of the forearm;*Y* rotation, as flexion of the shoulder and/or the elbow and;*Z* rotation, as abduction of the shoulder and/or the elbow.

For the identification of limb position, the location of the forearm in space independently of the hand motion is needed. While *X* rotation reflects in the different hand motions (pronation/supination), *Y* and *Z* rotation provide more information about the limb’s pose in space. It is important to note that the absolute Euler angles are not only determined by upper limb position but also through further movement relative to the global reference frame. Motion in the XY plane (e.g., by walking through a room or rotating the upper body) occurs frequently during daily life activities. This reflects in *Z* rotation, making this measure dependent on multiple factors and unreliable as a unique parameter for identification of the desired limb pose.

Movement in the XZ-plane occurs seldom, since the subject needs to rotate the whole upper body into a horizontal position. This happens, for example, when the subject lays down. By ensuring that the subject is always standing or sitting up right, which is the natural position, further influence on the external *Y* rotation can be neglected. The absolute *Y* Euler angle can be mapped to the sum of shoulder and elbow flexion. Although a single IMU does not allow the derivation of individual elbow and shoulder angles, it indicates the orientation of the forearm in the sagittal plane. With a predefined selection of discrete limb pose states, as shown in Figure 2, the *Y* rotation can be used as pose identifier. Within this work, the summed flexion angle was sufficient for the selected limb poses (see Section 2.3). Nevertheless, the concept can be expanded for future work with a second sensor of orientation placed on the upper arm to correctly identify the separate shoulder and elbow angles [48].

#### 2.1.2. Performance Evaluation

To identify states in need of adaptation, the performance of the reference model was analyzed individually for each combination of hand motion and limb pose. Underrepresented states are likely to be similar to other classes because of a lack in distinguishable information. Sensitivity is an evaluation measure that quantifies how often a hand motion is actually predicted correctly, Equation (Equation 1). Underrepresentation of a class and consequent misclassification as a false negative directly reflects in lower sensitivity. Therefore, sensitivity was used as a measure for the representation of each combination of hand motion and limb pose within the reference classifier.
(1)$$Sensitivity=\frac{True\;Positives}{True\;Positives+False\;Negatives}$$

The goal of the adaptive learner was to overcome underrepresentation of the states by adapting the classifiers training set. To avoid an additional calibration, new sample points were acquired during active usage of the virtual prostheses, i.e., without ground truth labels. Labels for the new sample points were provided by classification using the previous reference model. To avoid the addition of false data points, the precision was included as a further measure. Precision gives an indication of the number of correct labels, using the following equation:(2)$$Precision=\frac{True\;Positives}{True\;Positives+False\;Positives}$$

The identification of underrepresented states based on sensitivity and precision relies on the specification of thresholds.

Previous work assumed 90% overall accuracy as good performance [12]. In this work, this quantification was conservatively translated into a desired sensitivity of 90%. In application, this threshold aims for 9 out of 10 correct executions of a specific state. Setting the threshold for the sensitivity came at the benefit that the class-specific measure was independent of the total number of classes, since true negatives from all other classes were not included. For the minimum precision of a classifier for a certain state, the same measure was assumed. With a precision threshold of 90%, a maximum of 1 out of 10 identified samples should be incorrect. The equal sensitivity and precision thresholds are expected to lead to a balanced model, where underrepresented classes are more likely to be identified but misclassifications are reduced. Both threshold values can be seen as hyper-parameters (Table 1) which can be adjusted in further studies. Within this work, the values presented in Table 1 were selected to investigate the feasibility of the algorithm.

#### 2.1.3. Adaptation

To augment the training set of the reference model in correspondence with previously identified underrepresented states, datasets recorded during everyday use were analyzed. These datasets were recorded during usage of a prosthesis with the previously trained reference classifier. Hand motion and pose labels were generated for each feature vector and sorted into a matrix structure, as shown in Figure 1. A comparison between the identified underrepresented states and the matrix indicated whether data points were available for the classifiers adaptation.

Since the hand motion labels of the available sample points were dependent on the previous reference model and no ground truth was available, this data may have been mislabeled. Adding falsely labeled data points in the calibration set would hinder the model learning the correct class boundaries and directly influence its performance. Therefore, a further post-processing method was introduced to minimize the addition of wrongly linked data points. The post-processing method was inspired by majority voting, which is a control algorithm applied to increase the performances of classifiers [49].

Majority voting takes into account a number of previous samples ($n_{MajVoting}$) and outputs the class that is mostly represented within the window. By analyzing a number of samples, a new hand motion needs to be predicted multiple times before being executed, thereby avoiding sudden changes [50]. However, with an increased number of samples analyzed, the required recording time window is extended. In a case of majority voting for a real-time application, the length of the decision window is limited by the time of acceptable command-actuation delay. Although majority voting was only applied as post-processing method on a previously recorded dataset, a transition to real-time applications is desirable. A maximum acceptable delay of 300 ms was identified by [22]. To stay below this threshold and maintain a buffer for processing time, a delay of 250 ms was determined (corresponding to two samples with a window length of 200 ms with 50 ms overlap). The majority vote sample number was calculated with the formula $n_{MajVoting}=2m+1$ [22], whereas *m* reflects the number of samples within the delay time. Considering the previously defined number of samples (m=2), $n_{MajVoting}$ was set to 5.

After post-processing the samples for each state, desired states were adapted. The training data of the initial calibration model were extended by the sample points and the classifier retrained, resulting in the adapted model.

### 2.2. Experimental Validation

#### 2.2.1. Participants

Ten able-bodied subjects (6 males, 4 females, mean age ± 29) participated in the experimental study after giving written informed consent. All participants performed the experiment with their dominant (right) hands. The subjects had no knowledge of the experiment’s intent and did not have any proficiency with myoelectric control. Demographic details of the participants are shown in Table 2.

#### 2.2.2. Data Collection

A Myo armband (Thalmic Labs, Burlington, VT, USA) was used as the EMG data acquisition system (Figure 3). It consists of eight circularly arranged sEMG electrodes and an IMU sensor [51]. EMG signals are amplified and provided in an 8 bit range (−128 to 127), and a 50 Hz notch filter is applied internally to remove power-line interferences [52]. The Myo Armband samples EMG and IMU data at frequencies of 200 and 50 Hz, respectively [51]. A USB adapter enabled wireless Bluetooth communication between the device and associated laptop which housed a 2.1-GHz processor and 8 GB of RAM. The Myo Armband’s availability and affordability [52] enforced its selection as acquisition system for this preliminary analysis.

The forearms of the subjects were shaved and cleaned with disinfectant (70% alcohol) before sensor placement. The Myo Armband was slid onto the forearm in a predefined pose (90${}^{\circ }$ flexion, no elbow flexion, palm facing inward) with the logo (marking electrode 1) pointing upwards. The device was positioned at the thickest part of the forearm. The tightness of the armband was adjusted with clips in between the electrodes to ensure maximum electrode-skin contact for all channels. Clips were attached in pairs at opposite links to ensure symmetric electrode placement. The total forearm length (midpoint of the lateral and medial epicondyle to the radial styloid) and the relative device position would allow intra-subject comparison. Overall, the relative Myo Armband position ranged from 10 to 25% of the forearm length.

The open source software tool BioPatRec [53] was used for the experimental validation. It allows EMG signal recordings, signal treatment, feature extraction, pattern recognition and control in a modular manner, as schematically shown in Figure 4. Each module enables the selection of user-specific parameters. The additional integration of IMU data was part of this investigation. For this purpose, two modules—IMU recording and IMU feature extraction—were extended using IMU information (highlighted in blue in Figure 4). A detailed description given in Appendix A.

BioPatRec’s integrated virtual interface Integrum VRE was used to visualize the real-time output of the classifier, and the TAC test [47] was adopted to quantify the classifier’s performance. The TAC test requires the user to move a virtual prosthesis (gray hand in Figure 5) into a reference hand (shown in green in Figure 5). The virtual prosthesis is controlled using actuation commands derived from the real-time classification. Each hand motion label is related to specific DOF of the virtual hand. Every active hand motion classification leads to movement in the predefined direction. Prediction of the rest class (*r*) corresponds to no movement of the virtual hand prostheses. The TAC test includes the following measures:Completion rate, the relative amount of completed target motions;Completion time, the time required to reach the target position;Selection time, the time until the desired target hand motion is classified for the first time.

In the presented experimental setup, the target displacement was conducted in one DOF. This allowed completion of the task with one hand only and derivation of hand motion-specific evaluation measures.

Similarly to prosthesis application in every day life activities, a misclassified or unintended hand motion requires an active antagonist movement. The time required for the correction of the unintended hand motion directly reflects on the evaluation measures of the TAC test, making it a more realistic performance measure than offline evaluation [47].

The level of difficulty of the TAC test can be varied by adjusting the amount of displacement of the virtual prosthesis from the initial position (distance) and allowance around the target position (range). The target position needs to be maintained for a specified amount of time (dwell time) and completed within a maximum time frame (trial timeout) [47]. The distance (D) and allowance (W) can be summarized as ID, as described by Equation (Equation 3) [54], to allow comparability of multiple tests.
(3)$$ID=log_{2}\left(\frac{D}{W}+1\right)$$

Only one variant of TAC test parameters was applied because previous studies reported no significant linear relation of the ID and evaluation measures [54]. The target distance was set to 60${}^{\circ }$ with an allowance of 8${}^{\circ }$, corresponding to an ID of 3.08 (previous studies ranged between 1.8 and 4 [46,47,54,55]). Similarly to these studies, the dwell time and timeout time were set to 1 and 15 s, respectively.

For each subject, recordings with six different hand motions, three limb poses, and three external load variations were conducted. Hand motion classes included the most common degrees of freedom—open-hand (oh), closed-hand (ch), wrist pronation (wp), wrist-supination (ws), [46,56,57], pinch grip (pg) and rest (*r*), as shown in Figure 6. Limb poses were restricted to the sagittal plane and included higher shoulder flexion tasks, such that gravitational force related effects were modeled [34]. As shown in Figure 7, three limb poses were included in this study: 90 degree elbow flexion (central picture) and two extreme positions (left and right pictures). The latter represent arm positions with less comfort and large variations from the initial position, which leads to an increased limb position effect [34]. Three variations of external loads were used for this experimental evaluation, to simulate no weight (0 g), the weight of a light-weight prosthesis (400 g) and the weight of the prosthesis while holding a standard object (600 g) [29]. A glove with velcro strips was worn by the subject on his/her dominant hand. The glove consisted of compartments where additional weights could be attached for subsequent measurements. A mark on the ground indicated subjects to stand 1.5 m in front of the monitor, which displayed actuation commands to the user. An example of the experimental setup is presented in Figure 8.

### 2.3. Experimental Protocol

After the sensor placement and the preparation of the subject, the experimental protocol (Figure 9) included three phases:*Initial calibration*, conducted dynamically. Figure 10 shows the order of instructed limb poses per hand motion. A video of the dynamic routine covering the identified limb poses guided the participants to reach and maintain a specific hand motion. Each hand motion was conducted continuously for 10 s, followed by a 5 s rest period. Within the hand motion recording, limb position instructions changed every 2 s. The aim was to record 2 s in each of the three predefined limb positions. To account for the subject’s reaction time, pose 1 was added in front and the end of the hand motion recording. Only the middle 6 s of the recording were stored as calibration dataset and labeled with the instructed hand motion. This procedure was repeated three times for all six hand motions, resulting in 18 s of labeled data per class. The recorded dataset was processed and used to train a baseline model.*Familiarization*, where the subject was familiarized with the virtual environment by performing multiple series of TAC tests (one series is defined as five target displacements covering all predefined hand motions, excluding rest, in randomized order). Within the familiarization phase, external loads were added to simulate an unknown confounding factor during daily usage. EMG and IMU data were recorded and used to selectively adapt the calibration dataset (Section 2.1). Figure 11 shows the different variations of limb position and external loads applied. For each combination of limb position and external load, a series of TAC tests covering all hand motions was conducted. This resulted in nine repetitions of five targets with a maximum duration (timeout) of 15 s each, leading to a maximum total duration of 11 min and 15 s. At the end of the familiarization, a new classifier was trained based on the new calibration set. Before starting the testing phase, a five minute break was included to avoid fatigue.*Testing*, where the baseline model and an adapted model were tested using the TAC test. Each hand motion was tested for both classifiers multiple times, and each variation in limb pose and external load was covered. Figure 12 presents the combination of TAC test series conducted during the testing phase. Each classifier was tested three times, in randomized alternating order. After completing a TAC series with both algorithms, the test recording was paused and a different variation of external load was applied. The order of weight variations, limb poses and hand motions was randomized for each pair of TAC series. The maximum total recording time for the testing phase was 7 min and 30 s (six tests with a maximum completion time of 75 s).

### 2.4. Data Processing

A segment length of 200 ms at a time increment of 50 ms was chosen, similarly to parameters used in previous work [46,58]. This resulted in a set of segments per channel, where the total number of segments and the number of data points per segment depend on the recording time and the sampling frequency, respectively.

In the subsequent step, relevant features were extracted from the already segmented signal to reduce the amount of information for classification. A combination of four EMG features [53,59], namely, mean absolute value (mav), zero crossing (zc), slope sign changes (ssc) and waveform length (wl) [60], was adapted for this study.

To facilitate the classification of hand motions such as wrist rotation (e.g., pronation/supination), the absolute Euler angle *X* was included as an IMU feature. As depicted in Figure 3, the absolute *X* rotation within the reference system reflects the orientation of the forearm. Euler angles around *Y* and *Z* axes were not included, since they can be influenced by other movements independent of the upper limb (e.g., upper body rotation or relocation in space) and do not provide additional information for this study. Since linear and rotational acceleration values only changed when the device was in motion (e.g., during dynamic movements), they did not provide additional value during static hand motion. Due to the lack of additional information and sensitivity to noise, they were also excluded as measures for wrist rotation. This led to four EMG features for each of the eight EMG channels and one overall IMU feature, which resulted in 33 components per window.

The linear discriminant analysis (LDA) model, a supervised classifier, required an initial calibration set to learn the class boundaries, and therefore needed a feature set with known label. A calibration set was recorded by instructing the subject to conduct certain hand movements. Then the commands were mapped to generate a feature set with ground truth labels. This calibration set was randomly split into a training and a test set. After generating a LDA model with the training data, it was evaluated based on the test set.

The randomized split of the calibration set led to different models, which resulted in varying performance among subjects. To reduce the statistical variation, a 3-fold method was applied. In other words, the splitting, training and testing procedure was repeated three times. The classifier most able to identify hand motions correctly (sensitivity) was then selected as the reference model.

While previous data processing steps were conducted in Matlab and BioPatRec [53], post-processing for the visualization of sample point distribution was handled in Python using the machine learning library scikit-learn [61]. To allow a more interpretable visualization of class distribution for each subject, the number of features per sample point (originally 33 in the real-time classification model) needed to be reduced. The machine learning library scikit-learn [61] facilitated the feature reduction through its linear discriminant analysis class. First, an LDA classification model was generated for each subject using the *fit* function. Second, the number of features per sample point was reduced by projecting the initial feature points onto a new linear subspace using the *transform* function. To allow visualization in a two-dimensional plot, two new arbitrary dimensions maximizing class separation were identified. For each subject, all recorded sample points were recalculated into the new feature space and plotted for all hand motion classes. Although feature reduction comes at the cost of information loss, the new dimensionality preserved the main discriminative information (covariance represented in the reduced feature space ranged from 66 to 90%) and allowed an initial analysis of sample point distribution.

## 3. Results

### 3.1. Limb Pose Estimation

For each available hand motion feature set, the corresponding *Y* Euler angle was analyzed. Figure 13 depicts the *Y* Euler angle for each sample point of a recorded hand motion sequence (in the example, data from subject 10). Calibration sequences are shown for six different hand motions (colored lines). Calibration was conducted dynamically, covering three positions (see Figure 2) multiple times (see Section 2.3). The dynamic movement is represented in the variation of the *Y* Euler angle in Figure 13. Thresholds (visualized as red dashed lines and quantified in Table 3) indicate the corresponding position for each Euler *Y* angle and allow derivation of the limb position label. By saving the pose labels in the same structure as the hand motions, the two labels can be linked during further processing.

### 3.2. Precision and Sensitivity

A low class-specific precision implies that samples from other classes were frequently misclassified as the analyzed hand motion (false positives). A combination of a low sensitivity score and a high precision score can identify states which lacked in detection rate, but if they were detected, the label is likely to be correct. These states were identified as potential states to be adapted with sample points generated based on the reference model.

To derive the state-specific evaluation measures, the predicted hand motion labels and their ground truth were separated by limb pose. Samples were stored in a matrix, whereas their predicted values and ground truth correspond to the row and column, respectively. Each row of the matrix was normalized by division by the total number of the corresponding hand motion and stored in percentages, resulting in a confusion matrix as visualized in Figure 14a for subject 10. The confusion matrices are used to derive the required parameters for the calculation of sensitivity and precision in accordance to Equations (Equation 1) and (Equation 2). True positives correspond to the diagonal values. With each true positive classified label, true negatives are ascribed to all other classes. False positives and false negatives reflect in the sums of non-diagonal values within the rows and the columns.

Figure 14b shows the result of sensitivity and precision per state and for all hand motions within a limb pose (last row). In this specific example, it is notable that hand motions ch and oh within pose 1 lack in sensitivity. The precision for these states was close to 100%, indicating the correctness of the labels predicted for these states. Low precision for *r*, wp and ws implies that samples are likely to be mislabeled as these classes. In case of subject 10, the predefined thresholds for sensitivity and precision lead to the identification of oh, ch and pg in pose 1 and *r* in pose 3 as states that can be adapted with an unsupervised data set.

### 3.3. TAC Test Results—MAIN Evaluation Measures

The overall completion rate, completion time and selection time were analyzed for all participants, and presented in Figure 15. The median completion rates for the baseline algorithm (BA) and adaptive algorithm (AA) were 53.33% and 56.67%, respectively. Individual classification rate varied between 33.33% and 86.67% for the BA, and from 40% to 66.67% for the AA.

For the baseline and adapted algorithms, median completion times of 3 and 4 s were recorded, respectively. Individual completion time ranged from 2.46 to 5.42 s for the baseline and from 2.44 and 5.49 s for the adapted model. It has to be noted that the completion time measure only included trials that were completed successfully within the timeout time of 15 s. Target positions reached only at the end of the test period reflect increased completion time.

The median selection times for both algorithms were 0.66 s. The minimum selection time achieved for both classifiers was 0.32 s, whereas the maximum times required to predict the desired motion for the first time were 1.27 and 1.89 for the baseline and adapted models, respectively.

### 3.4. TAC Test Results—Hand Motion Specific

A hand motion-specific analysis of the completion rate is shown in Figure 16. For the oh, wp and pg classes, the median completion rates (66.67%, 0% and 100%) over all subjects were identical for both classifiers. For ws, the adaptive algorithm led to a higher median completion rate compared to the baseline algorithm (33.33% versus 0%). Individual performance showed a completion rate of 100% with the adaptive algorithm, whereas the baseline algorithm recorded 66.66% maximally. The class-specific median completion rate for ch of the baseline algorithm was higher than that of the adaptive algorithm (100% versus 83.33%), but individual completion rates of both ranged from 66.66% to 100%.

### 3.5. TAC Test Results—Subject Specific

Since calibration set adaptation varied for each subject (Section 3.6), subject-specific completion rates were identified for each hand motion and overall; see Figure 17. The adaptive classifier led to increased classification rates for subjects 1, 5 and 7 (+6.67%, +6.67% and +13.33%). In four cases (subjects 2, 4, 8 and 9) no difference was recorded for the overall completion rate. For subjects 3, 6 and 10, less TAC trials were completed with the adaptive algorithm (−6.67%, −6.67% and −26.67%). It is worth noting that in multiple cases, hand motions were completed with the adapted algorithm, although the baseline did not result in a successful trial at all. In the case of subject 7, the oh completion rate increased from 0% to 66% through adaption. ws improved from 0% to 33% for subjects 1, 5 and 8. Performance reduction resulting in 0% for the adaptive algorithm was observed for wp of subject 3, whereas the completion rate of ch increased.

### 3.6. Adaptation—Overall

The aim of the adapted algorithm is to increase calibration dataset to improve the overall robustness to different external conditions. To understand the the effectiveness of the proposed method, the number of feature vectors (also referred to as sample points) adapted was identified. Over all subjects, 13,186 data points were added from the familiarization phase. This corresponds to 82% of the initial data size for each subject. Table 4 shows that 41% of the data were recorded in limb position 2, 32% in position 1 and 26% in position 3. The numbers of data points recorded under the influence of external loads were similar for each condition (31% for 0 g, 35% for 400 g and 34% for 600 g), as shown in Table 5.

More deviation was observed in the amount of hand motion samples added (see Table 6). Considering all subjects, relatively few samples were added for pg (2% samples) and ch (10%), and a larger number of additional samples were included in the wp, oh and *r* classes (26%, 25% and 17%). In reference to the hand motion-specific completion rate, Section 3.4, classes with little adaptation (pg and ch) generally presented similar performances with both algorithms (median completion rate of 100%). The high performance with the baseline could indicate that the class was identified as well represented, and therefore did not require additional samples. wp, oh and *r* were less frequently completed with the baseline algorithm (median completion rate ranging between 0% and 66.66%). These classes were also more frequently identified as underrepresented, leading to an increased amount of adaption of the corresponding hand motions.

### 3.7. Adaptation—Subject Specific

Since different combinations of hand motions and limb positions are identified as underrepresented for each subject, the overall amount of data points does not reflect the individual adaptation. Therefore, added data points were identified for each subject specifically. Figure 18a shows the amount of different hand motion samples that were added on an individual basis, and Figure 18b,c presents the same analysis for the limb positions and external loads. The number of adapted sample points per subject ranged between 248 and 2778, with an average of 1386 data points. The initial amount of data points of each calibration set was 1602 and is highlighted with a dashed red line in Figure 18. In contrast to hand motion and limb position variations, external load variations were not dependent on the previous performance evaluation. Figure 18c shows in the case of subject 5 that sample points recorded under two added loads were added, even though only one combination of limb pose and hand motion was adapted.

Reduced feature maps were generated to visualize data points used for model generation. Therefore, the 33 features of each sample point were reduced to two main components (identified by the LDA model) and plotted with each dimension on one axis. Figure 19 shows the reduced feature map for subject 1. The feature map is plotted for the sample points of the initial calibration set used for the baseline classifier. On the right side, a feature map with the adapted points (circled in black) is visualized. For subject 1, it is possible to note that the calibration dataset of the oh and ch class was expanded. As also seen in the bar plot for subject 1 (Figure 18a), mostly oh data points were added.

## 4. Discussion and Conclusions

### 4.1. General Performance

The overall TAC test results indicate a higher median completion rate for the adaptive algorithm. The results show inter-subject variance in completion rates of the assigned tasks, particularly for the baseline measures. However, the proposed algorithm reduced the performance variation, i.e., between best and worst individual results from 50.33% to 26.67%. Similar findings while using the TAC test have also been reported in [54]

Performance has been shown to be dependent on the type and duration of hand motions within a dataset [62]. The increased variation of the baseline algorithm leads to the assumption that the quality of initial calibration data is affected by several subject-related conditions, e.g., by variation during the calibration phase. This highlights the importance of an adaptive method which has shown to improve the initial calibration set according to the user’s requirements and features. This is detrimental for subjects with limb loss, where the inter-subject variation has found to be higher [43].

### 4.2. Hand Motion Performance

It is worth noting that for both algorithms, hand motion-specific performance varied largely. While classes such as ch and pg mainly performed higher than the overall median, ws and wp led to lower results. An analysis of the adapted data points showed that these hand motions were among the most adapted classes as well. These findings indicate that the class boundaries for wp and ws were not distinguishable clearly. This could be related to the limitations of the sensor system used for this preliminary evaluation, or difficulty in distinguishing wp and ws with no grip pattern due to the lack of sufficient muscle activation [63], since lack of muscle recruitment makes it difficult to separate the wrist rotations from the rest class. Relatively large adaptation of the *r* class supports this statement, since this indicates a low sensitivity of the hand motion.

### 4.3. The Influence of External Loads

Table 5 shows that all variations of external loads were adapted, although they were not included in the initial calibration model. Even if only one combination of hand motion and limb position needed to be adapted, the source for new data points was not limited to the known external load of 0 g, but included samples under the influence of 400 and 600 g, as can be seen Figure 18c for subject 5 and 9. In the case of subject 5, the adaptation of unseen conditions even improved the overall TAC test performance (Figure 17). Since added weights induce increased muscle recruitment [64], the results imply that the model was able to cope with increased muscle activation. This is reflected in the magnitude of the features and may be the reason for the improved performance of hand motions, which originally require little muscle activation under external load. In particular, the improved ws completion rate (Figure 16) supports the theory that the inclusion of external loads leads to increased separability.

### 4.4. Quality and Quantity of Adaptation Set

A subject-specific analysis of the changes induced by adaptation was inevitable because myoelectric patterns differ between subjects [65,66]. Figure 18a shows that different hand motions were adapted for each subject. However, there is no visible trend regarding the amount of additional data and subject-specific performance improvement. In some cases, adapted sample points did not necessarily lead to better subject performance with the adaptive algorithm. It can therefore be argued that the amount of data was not the driving factor for successful adaptation. This observation demonstrates that qualitative data is also an important criterion in conjunction with a greater number of data points [10].

The desired quality of adapted data points led previous studies to conclude that supervised calibration sets with high-confidence labels are more feasible than unsupervised [35,43]. This study did not require a supervised dataset for adaptation, but nevertheless shows relative performance improvements for three subjects (1, 5, 7). The analysis of feature map representations helps to explain that in these cases the new data points resulted in a distinct separation between the classes. Analysis of the feature maps with decreased performance shows no improved class separability. Added sample points rather led to overlap of class boundaries. It has to be noted that baseline performance of the corresponding subject was high (86.67%). In this case, added data points might have been of high confidence with the potential to over-fit the model with excessive and/or unnecessary amounts of data [35]. Subject-dependent success of adaptation was not only observed with the adaptive algorithm, but also be seen for supervised methods [43]. The decreased class separability of multi-limb pose classifiers has previously been shown to influence class boundaries [33].

Adaptation of the developed algorithm is dependent on labels from the previous model. Consequently, all adapted feature maps show that the algorithm picks up data points close to the initial baseline mean. This is similar to adaptation proposed in [35], where only data points close to the initial class mean were added, but comes with the benefit of unsupervised calibration. The slow shift of the mean allows the model to adapt to slow changes, which is relevant for other factors such as muscular fatigue and variations in skin impedance [65].

### 4.5. Evaluation Measures

Reviews of adaptive algorithm studies have shown that frequently, usage of classification accuracy as an evaluation measure [45,67,68,69,70] includes a virtual testing environment that reflects misclassification during the completion time, but neglects the need for correction of unintended movement. In this study, analysis of the offline classification accuracy on the testing set from the initial calibration showed lower results for the adapted algorithm, but with no significant difference (*p* < 0.05). These testing results only take into account the initial calibration testing set, where similar performance was expected. Evaluation on new data points in different conditions is more important and represented in the online evaluation measures. Contrary to the offline classification accuracy, the results of the online evaluation, which include adverse effects of misclassification, are promising. In accordance to [71], these observations indicate that realistic test scenarios are crucial for the evaluation of clinical applications.

### 4.6. Study Limitations and Future Work

In this paper, an adaptive algorithm based on a pattern recognition classifier was introduced and evaluated in an experimental study, including ten able-bodied subjects. Multiple mitigation methods encountering performance decreases due to confounding factors have been presented in previous work. The proposed algorithm furthermore considers the clinical feasibility and includes the novel aspect of minimizing calibration time while covering variations of external influences. To the best of our knowledge, this is the first online study investigating unsupervised adaptation of pattern recognition controlled prostheses under the influence of multiple confounding factors. The presented adaptive model identified combinations of hand motion and limb positions where additional data points were required to ensure satisfactory performance. All available data points for these combinations were adapted. For a few subjects, this led to a considerable number of adapted sample points included in the initial calibration set. However, adaptation of a large amount of data points could limit the performance of the model in two ways:The amount of training data could influence the computation time of the algorithm. While the LDA itself is computationally efficient [72], a larger training dataset requires more time for the computation.Data included during the adaptation process may influence the class boundaries negatively. As mentioned in [35], an excessive amount of data is not necessarily beneficial because it may lead to over-fitting.

In the current model, new samples are not checked for information gain compared to the existing class distribution. This will be further investigated in future studies, where we will focus on improvements to and extensions of the proposed adaptive model. The identification of underrepresented states and the selection of adapted data are dependent on hyper-parameters. Optimization and introduction of new hyper-parameters in future applications will help to avoid over-fitting. Existing data limitation concepts, such as forgetting factor, deletion of previous data or information gain evaluation [50,73,74], could be integrated to encounter excessive adaptation.

The presented model used unsupervised data for adaptation; however, the labels needed to be generated for inclusion into the supervised classification model. The algorithm in this paper was able to classify hand motions within the familiarization and testing phase based on the initial calibration model, although it was unfamiliar with weight-induced variations. Since weight-induced variations were mainly reflected in muscle activation rather than variations in muscles recruited [64], the unseen samples were observed to be in areas close to the initial class, and the proposed model was capable of labeling these. Adapting this data to be close to the mean of the initial class provides sufficient adaptation for slow changes [35], but will neglect complete domain shifts if the calibration model is unfamiliar with them. Sudden changes in pattern and new class boundaries, such as induced by external factors as electrode shift or donning/doffing [9,65,75], are currently not identifiable by the current implementation and limit the application of the adaptive learner to slow changes. To generalize the model for further external influencing factors, hybrid concepts such as an independent label generator [70] will be explored. Data fusion has been shown capable of increasing model knowledge and may be used as additional information source for such parallel proxy models.

Finally, future studies will include a more extensive experimental protocol, with a larger number of participants (including subjects with and without limb loss), multiple sessions and the use of a sEMG sensor system certified for medical use. An extended protocol should provide more insights regarding the long-term performance of the adaptive algorithm and its potential translation in real-life applications.

## Figures and Tables

**Figure 1 sensors-21-07404-f001:**
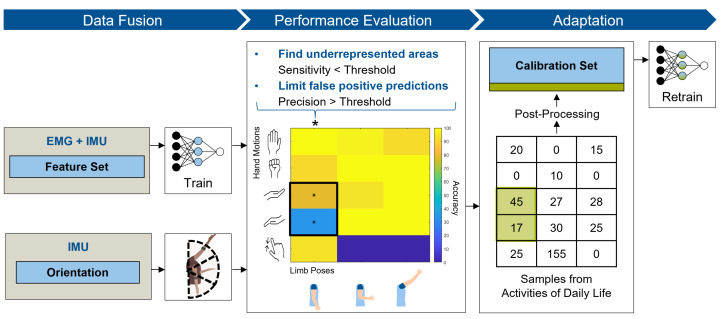
A schematic overview of the multi-modal adaptive algorithm. The method consists of 3 steps: (1) Data fusion: EMG and IMU data are fused to train the calibration model; pose estimation labels are generated and fused with the hand motion labels. (2) Performance evaluation: Sensitivity and precision are evaluated for each combination of hand motion and limb pose, to identify classes in need of adaptation (marked with a star). (3) Adaptation: A further unsupervised stage with labels generated by the reference classifier is taken into account. Sample points are sorted along hand motion and limb pose. The matrix is compared to the previously identified states and potential sample points are marked (marked in green). Post-processing is applied to the sample points to reduce the probability of adapting wrongly labeled data points. Adaptation of the calibration set is conducted and the model gets retrained.

**Figure 2 sensors-21-07404-f002:**
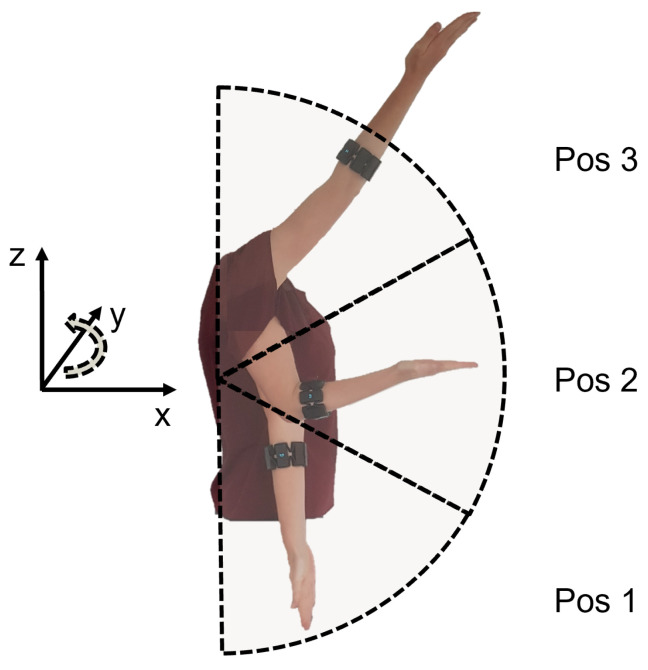
Pose estimation based on IMU data. Transformation of the IMU’s quaternions into a global reference system provides the rotation of the limb around a set *Y* axis. The *Y* rotation angle can be mapped to three predefined poses.

**Figure 3 sensors-21-07404-f003:**
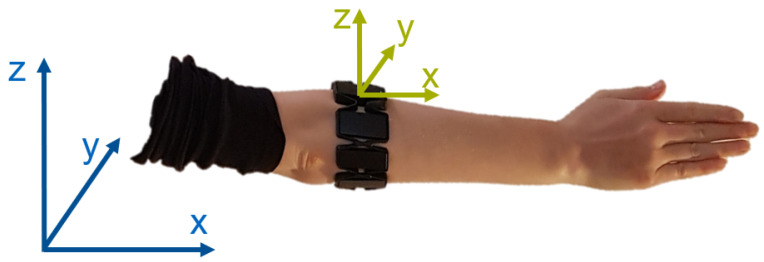
The Myo armband with a local body-fixed coordinate system (green) and global space-fixed reference axes (blue). Linear and angular acceleration are provided relative to the local system. Orientation information for quaternions is supplied with reference to the global axes. Euler transformation yields absolute angles in reference to the space-fixed axes.

**Figure 4 sensors-21-07404-f004:**
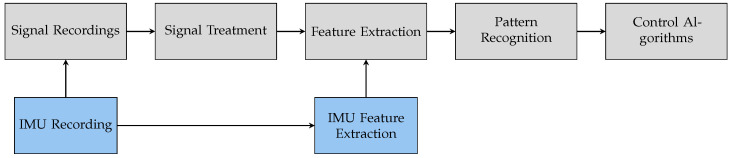
BioPatRec modules adapted from [53] and newly developed extensions for IMU processing (marked in gray and blue, respectively).

**Figure 5 sensors-21-07404-f005:**
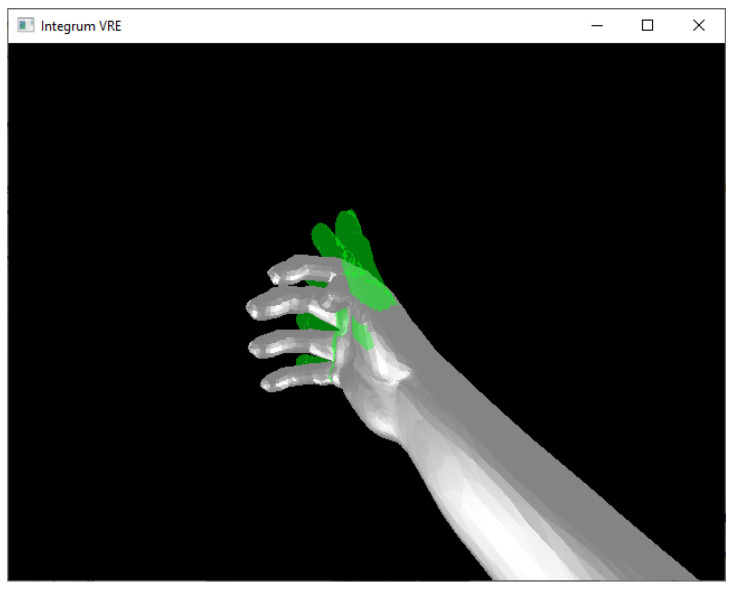
TAC test within the virtual hand environment provided by BioPatRec. The green hand indicates the reference hand; the gray hand represents the virtual prostheses. The participants could control the virtual prosthesis, and each hand motion was predicted in real-time.

**Figure 6 sensors-21-07404-f006:**
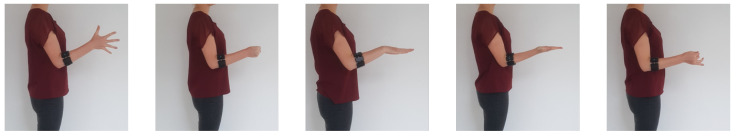
Active hand motions included in the experimental evaluation. From left to right: oh, ch, wp, ws and pg.

**Figure 7 sensors-21-07404-f007:**
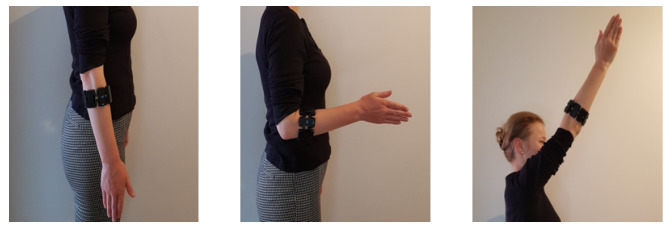
Limb poses included in the experimental protocol. Limb poses and corresponding angles from left to right: pose 1${}^{\circ }$—0${}^{\circ }$, shoulder and elbow flexion, pose 2${}^{\circ }$—90${}^{\circ }$, elbow flexion, pose 3${}^{\circ }$—135${}^{\circ }$, shoulder flexion.

**Figure 8 sensors-21-07404-f008:**
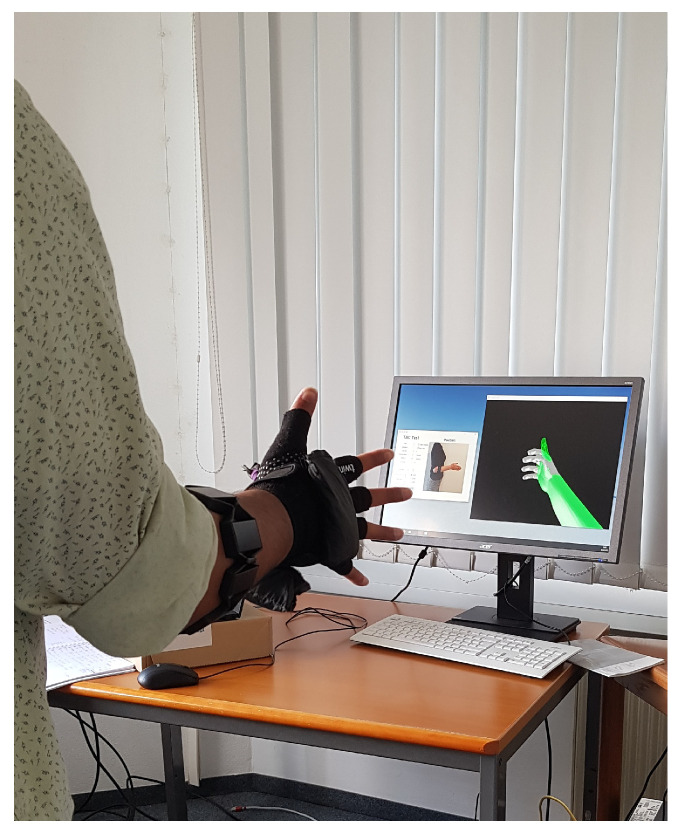
An example of a subject performing the TAC test.

**Figure 9 sensors-21-07404-f009:**
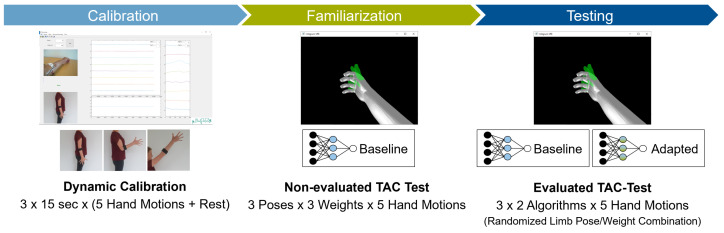
A summary of the experimental protocol.

**Figure 10 sensors-21-07404-f010:**
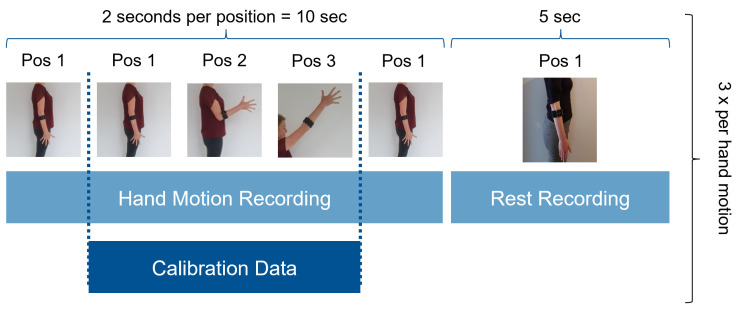
Calibration phase procedure. The subject was instructed to conduct a hand motion continuously for 10 s, followed by 5 s of rest. Within the 10 s, limb position commands appeared every 2 s. The first and last 20% of the hand motion recording were cut to account for reaction delay. This sequence was repeated three times for each of the six hand motions.

**Figure 11 sensors-21-07404-f011:**
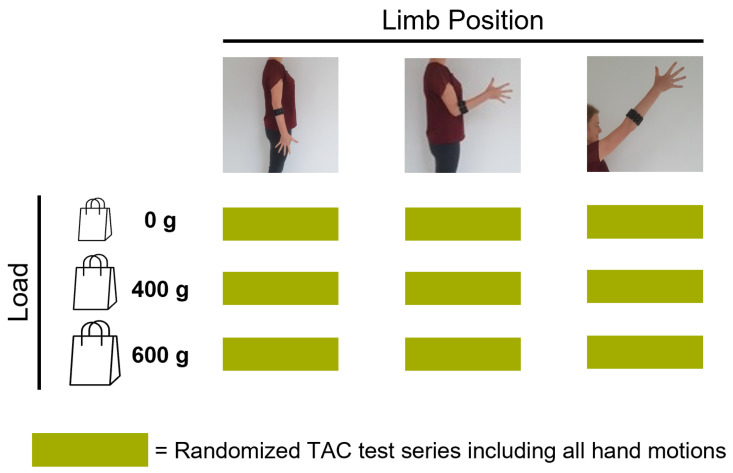
Familiarization phase procedure. The subject was familiarized with TAC test in for all variation of external factors. A TAC test series was conducted for each combination of limb position and external load.

**Figure 12 sensors-21-07404-f012:**
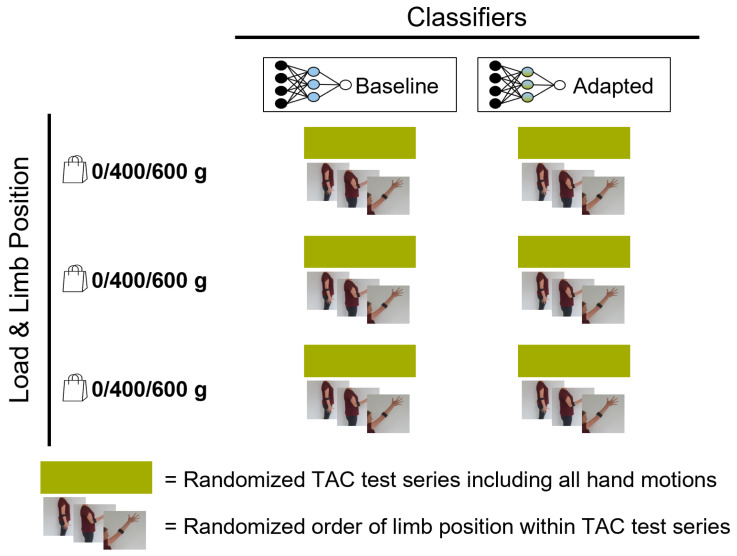
Testing phase procedure. Classifiers (baseline and adapted) were tested in TAC test series three times each, in randomized alternating order. The external load was changed before each pair of TAC test series. Limb position varied for each hand position within the series.

**Figure 13 sensors-21-07404-f013:**
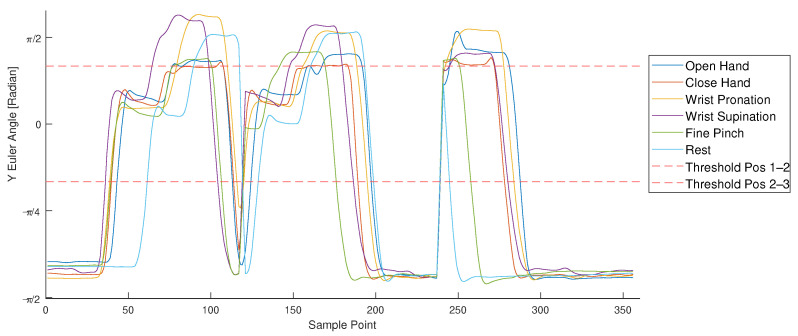
*Y* Euler angle and estimated limb pose labels recorded during calibration session for subject 10. Each colored line represents the recording of one hand’s motion during calibration. The *Y* Euler angle was plotted for each sample point and compared to predefined thresholds (red dashed lines), resulting in a limb pose label.

**Figure 14 sensors-21-07404-f014:**
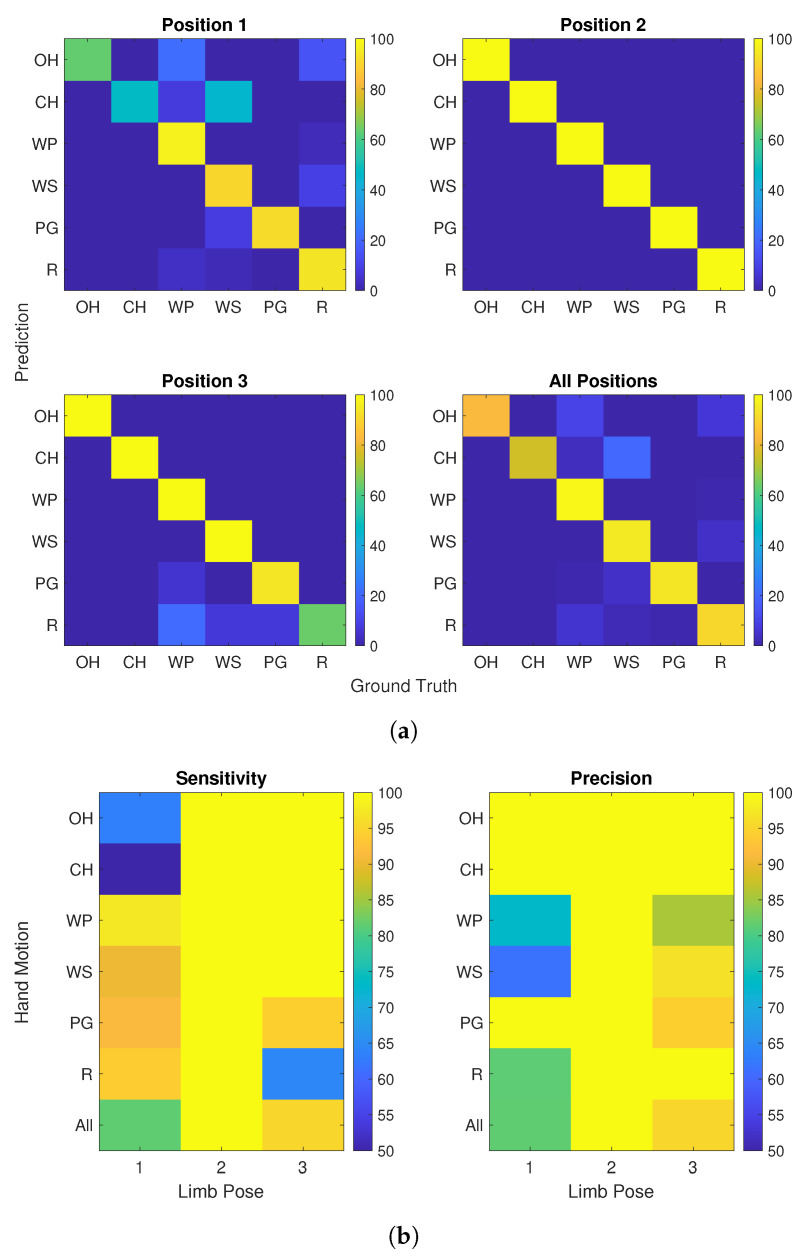
Performance evaluation separated by hand motion and limb position for subject 10 (after 3 k-fold). Sample classifier trained and evaluated with the calibration set including six different hand motions (oh, ch, wp, ws, pg, *r*) and three different limb poses (3:1 split). (**a**) Normalized confusion matrices per limb pose and overall, (**b**) Performance evaluation measures in % per hand motion (row) and limb pose (column).

**Figure 15 sensors-21-07404-f015:**
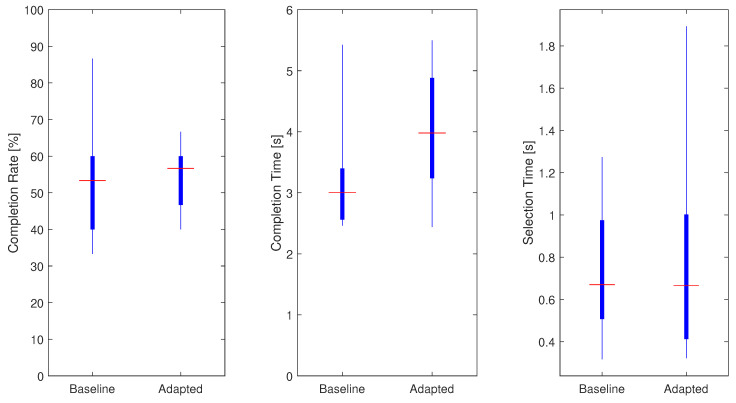
TAC test results summarized for all subjects, hand motions and external loads: completion rate (**left**), completion time (**center**), selection time (**right**).

**Figure 16 sensors-21-07404-f016:**
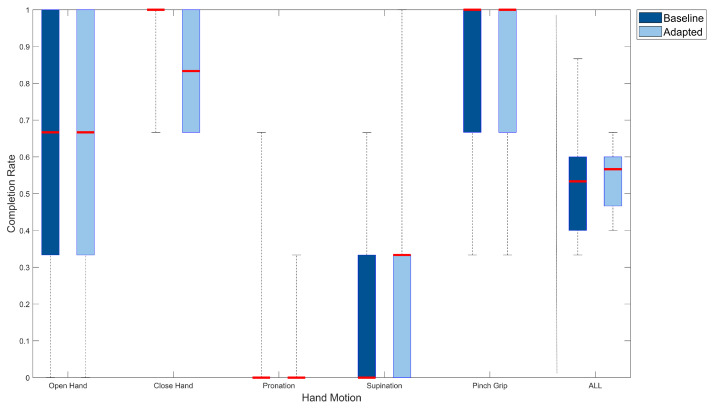
Hand motion-specific completion rate for all subjects and external loads. Box plots visualize median completion rate (red line), minimum and maximum completion rates (lower and upper end of whiskers) as well as the first and third quartile (lower and upper end of boxes).

**Figure 17 sensors-21-07404-f017:**
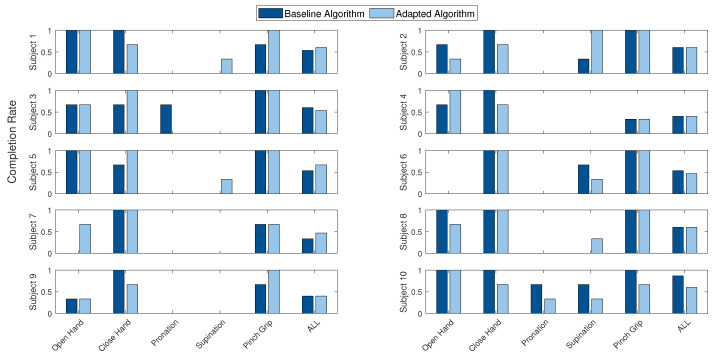
Hand motion-specific completion rate for each individual subject (including all external loads). From left to right: oh, ch, wp, ws and pg.

**Figure 18 sensors-21-07404-f018:**
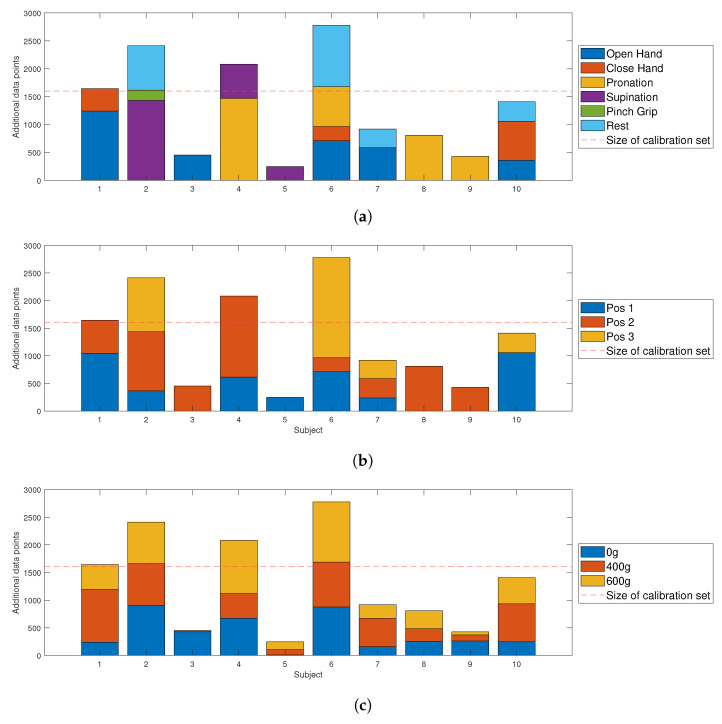
Amount and variation of sample points selected for calibration set adaptation. (**a**) Hand motion samples adapted per subject. (**b**) Limb position samples adapted per subject. (**c**) External load samples adapted per subject.

**Figure 19 sensors-21-07404-f019:**
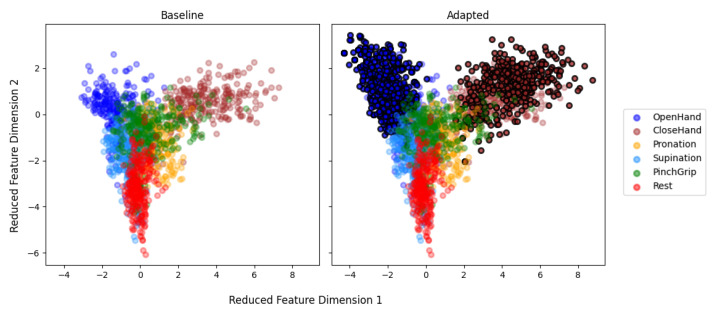
Reduced feature map before adaptation (**left**) and after (**right**) for subject 1. Adapted samples are marked with black circles.

**Table 1 sensors-21-07404-t001:** Hyperparameters for the adaptive learner.

Parameter	Abbreviation	Value
Minimum sensitivity	$sens_{min}$	90%
Minimum precision	$prec_{min}$	90%
Majority voting segment length	$n_{MajVoting}$	5 samples

**Table 2 sensors-21-07404-t002:** Subject-specific demographics and anatomic measures.

Subject	Age	Gender	Forearm Length (in cm)	Absolute Myo Position (in cm)	Relative Myo Position	Tightness (No. of Clips)
1	25	m	27	5.5	0.20	2
2	23	w	26	5	0.19	4
3	34	m	26	5.5	0.21	2
4	35	m	27	4	0.15	4
5	32	m	27	6.5	0.24	2
6	30	w	26	4.5	0.17	6
7	30	w	26.5	5	0.19	6
8	23	w	25	6	0.24	6
9	29	m	29	4	0.14	2
10	30	m	28	7	0.25	0

**Table 3 sensors-21-07404-t003:** Euler angle ranges for position estimation.

	Pos 1	Pos 2	Pos 3
Range	$x<\frac{-\pi }{3}$	$\frac{-\pi }{3}\leq x<\frac{+\pi }{3}$	$\frac{-\pi }{3}\leq x$

**Table 4 sensors-21-07404-t004:** Absolute and relative amount of adapted data points per limb position for all subjects.

Limb Pose	1	2	3	Total
Absolute [samples]	4278	5432	3476	13,186
Relative	0.32	0.41	0.26	1.0

**Table 5 sensors-21-07404-t005:** Absolute and relative amount of adapted data points per external load for all subjects.

External Load	0 g	400 g	600 g	Total
Absolute [samples]	4064	4628	4494	13,186
Relative	0.31	0.35	0.34	1.0

**Table 6 sensors-21-07404-t006:** Absolute and relative amount of adapted data points per hand motion for all subjects.

Hand Motion	oh	ch	wp	wp	pg	*r*	Total
Absolute [samples]	3350	1355	3424	2297	182	2578	13,186
Relative	0.25	0.10	0.26	0.17	0.02	0.2	1.0

## Data Availability

The data presented in this study are available online at this repository https://github.com/vjspi/biopatrec (accessed on 30 September 2021).

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
