# Peer review of "An Adaptive Multi-Modal Control Strategy to Attenuate the Limb Position Effect in Myoelectric Pattern Recognition"

_sensors, 2021, doi:10.3390/s21217404_

Round 1
Reviewer 1 Report
This study proposed a multimodal adaptive algorithm for hand motion recognition on healthy individuals based on data fusion of EMG + IMU. Three different arm positions and three loads were considered for the experiments to assess the robustness of the method. A Myo armband composed of eight sensors was used to record data sets. The authors conducted experiments on different configurations for an LDA classifier including calibration and adaptation. The methods are widely explained and detailed, accompanied by figures. Furthermore, the manuscript presents the improved reliability of the adaptive method analyzing the hand motions, poses, and weights.
Overall, the topic is interesting, and this paper presents diverse experiment results for demonstrating the effectiveness of the improvement with the proposed method. However, the presentation of the manuscript has room for improvement before its publication. The reviewer's comments are as follows.
Methods:
The states were defined as hand motions for each position of the hand. After reading the results, it was clearer that a classifier was trained for each individual and for each pose. However, no details about external load were presented. Please, include details in the Methods.
The features extracted from IMU were explained in the section Data Fusion. However, in Line 343, "IMU features (mean values of orientation, linear and angular acceleration)" are mentioned but it was unclear in Methods.
The terms for performance evaluation should be uniformed and reviewed. The equation of Sensitivity should be reviewed since True Negatives could be confused with False Negatives. Also, Table 1 includes specificity but it was not mentioned as a metric for the assessments. Specificity was only mentioned in Figure 1. Please, review if Precision is the correct metric is referred to throughout the text.
Lines 163 -167: For the majority vote method, it was mentioned that two samples can be included in this delay, defined as m. It was unclear if the buffer if 250 ms or 300 ms.
Please, include information about the formula used for the majority vote or add some reference.
Line 320: Why "four hand motions" were considered for calibration sets, instead of the six gestures (including resting)?
It is confused when used different terms when referring to the motion. In Line 321, extension and flexion are used instead of wrist pronation and supination. Figure 14 includes Side Grip but it was not mentioned in the manuscript. Please, use uniform terms. It should be reviewed throughout the text.
Line 338: Please, add to methods details of the technique used for reducing dimensional features.
Results:
The presented results from 3.1 to 3.6 did not specify if they were performed for a single participant or for all the participants. Also, it was not mentioned the specific weight to which it corresponds. Moreover, performance for different external loads was not included (besides of Table 5).
Section Results should include numerical results in the text and not only in Figures and Tables. Example: Line 346 (fewer misclassifications)
Line 385. Please, include more details about the normalization technique.
The methods should be in past temp. Example: Line 431 (Over all subjects, 13 186 data points WERE added from the familiarization phase). Please review that throughout the Results section.
Minor Concerns:
Figure 1. The abbreviations on Feature Set {C x F x W}t should be included in the legend.
Lines 199 - 200: The modules signal recording and feature extraction must agree both in the text as the Figure 4.
r Class is mentioned in Line 208 but it is defined in Line 231
Line 224: "id" and "ID" must agree both in the text as the equation 3.
Line 299: Please, review the sentence: "The absolute Euler angle is around X is derived".
Line 368: Please, review the sentence: "For each available hand motion feature set hand the corresponding Y Euler angle was analyzed".
Figure 19. Please review the term "Fine Pinch".
Reviewer 2 Report
- The text of the article must be corrected from the point of view of English grammar. For example, in line 148, the word "ad" appears instead of the word "as".
- More details on how the machine learning library (scikit-learn) was used should be given.
- Part 4 of this paper, it is suggested that the authors explain the workload and innovation in the text in detail.
